# Evaluation of the Efficacy of Pectoral Nerve-2 Block (PECS 2) in Breast Cancer Surgery

**DOI:** 10.3390/jpm13101430

**Published:** 2023-09-24

**Authors:** Jolanta Cylwik, Małgorzata Celińska-Spodar, Natalia Buda

**Affiliations:** 1Anesthesiology and Intensive Care Unit, Mazovia Regional Hospital, 08-110 Siedlce, Poland; jolacylwik@o2.pl; 2Anesthesiology and Intensive Care Unit, The National Institute of Cardiology, 04-628 Warsaw, Poland; 3Simulation Laboratory of Endoscopic and Minimally Invasive Techniques, Medical University of Gdansk, 80-211 Gdansk, Poland; 4Lung Transplant Department of Cardio Surgery Clinic, University Clinical Center in Gdansk, 80-211 Gdansk, Poland

**Keywords:** breast cancer, breast surgery, PECS II, intercostal nerve blocks, postoperative pain, mastectomy, oncological surgery, mammary gland, patient satisfaction, regional anesthesia

## Abstract

This study aimed to evaluate the efficacy of the pectoral nerves interfacial plane block (PECS II) in breast cancer surgery focusing on postoperative pain management and patient satisfaction. A prospective study was conducted, including 200 patients scheduled for breast cancer surgery. The participants were randomly assigned to the PECS II block and control groups. The PECS II block group received a preoperative interfascial plane block, while the control group received standard analgesia. Postoperative pain scores at 4 h intervals for the first 3 postoperative days, as well as opioid consumption and patient-reported satisfaction, were measured and compared between both groups. The PECS II block group demonstrated significantly lower postoperative pain scores at all measured time points (*p* < 0.001). Additionally, the PECS II block group showed reduced opioid consumption (*p* < 0.001), reported higher levels of patient satisfaction compared to the control group, and had a notably shorter stay in the postoperative care unit (*p* < 0.001). Integrating the PECS block with general anesthesia in breast cancer surgeries enhances pain management, reduces opioid use, and shorten postanesthesia care unit stay. The evident benefits suggest PECS as a potential standard in breast surgeries. Future research should further investigate its long-term impacts and broader applications.

## 1. Introduction

Breast cancer remains the most prevalent cancer among women globally. As its incidence increases by an approximate 0.5% yearly, so does the demand for breast cancer surgeries [1]. This upward trend underscores the necessity for optimizing both patient outcomes and the efficiency of healthcare resources.

Postoperative recovery plays a pivotal role in the overall treatment outcome of breast cancer patients. Not only does it influence the patient’s comfort and overall satisfaction, but it also has significant ramifications on the operational efficiency of the healthcare system, notably in the operating room (OR) and the duration of postanesthesia care unit (PACU) stays.

While general anesthesia remains the mainstay, the introduction of ultrasound in the OR has paved the way for more accurate and safer regional blocks as adjuncts. The potential benefits of adding regional blocks to general anesthesia, such as the Pectoral Nerve Blocks I and II (PECS I and II), extend beyond just pain relief. They might lead to reduced opioid consumption, thereby potentially decreasing opioid-related complications such as nausea, respiratory issues, and even increased mortality in cancer patients, as highlighted by recent research [2,3].

PECS I and II, compartmental blocks of the chest wall, are gaining traction as simpler and safer alternatives to the paravertebral block (PVB) [4,5]. However, there exists a gap in the current literature regarding the consistent benefits of integrating these blocks into routine breast surgery anesthesia. Many still debate the real-world analgesic advantages of the PECS II [6].

In light of this, our study delves deep into evaluating the efficacy of adding the PECS I and II to general anesthesia during breast cancer surgeries. We seek to elucidate its impact on postoperative pain, opioid use, OR efficiency, and postoperative stays, all in comparison to the standard general anesthesia regimen.

## 2. Materials and Methods

### 2.1. Study Design

This prospective study was conducted at the Mazovia Regional Hospital, a tertiary oncologic center, from January 2020 to December 2021, involving a cohort of 200 consecutive patients scheduled for breast cancer surgery. Inclusion criteria comprised patients who met the following conditions:

Age: Participants had to be older than 18 years of age.

ASA Classification: Patients classified under the American Society of Anesthesiologists Physical Status Classification System (ASA) with a status of 1, 2, 3, or 4 were eligible for inclusion.

Surgical Procedure: Eligible participants were those scheduled for breast cancer surgery.

Informed Consent: Patients who provided informed consent for participation in the study were qualified for inclusion.

The study’s design was prospective, with random allocation to either group. The surgical team was blinded to the performance of the PECS II, ensuring an objective evaluation of postoperative outcomes.

Group 1: Patients in this group underwent the surgical procedure under general anesthesia alone.

Group 2: Patients in this group received general anesthesia with an adjunctive interfascial plane block—PECS II.

Approval for this study was procured from the Bioethics Committee at the Regional Chamber of Physicians and Dentists in Warsaw under the reference number KB/1260/19.

### 2.2. Perioperative Management

Perioperative preparation time and procedures for both study groups were consistent. Upon OR admission, patients were monitored, including continuous ECG recording, intermittent non-invasive blood pressure measurement, and continuous transcutaneous oxygen saturation. Anesthesia induction and maintenance protocols were uniform across both groups, with the only distinction being the addition of the PECS II block for Group 2. After typical preoxygenation, intravenous induction of general anesthesia was performed. Patients received fentanyl (2 mcg/kg), propofol (1.5–2.5 mg/kg, individually adjusted based on age and coexisting conditions), and rocuronium (0.6 mg/kg) for muscle relaxation. Atropine was administered during induction if indicated. Anesthesia maintenance was achieved using desflurane (MAC 1.0–1.1) and fractional doses of rocuronium and fentanyl. Following general anesthesia induction, each patient in group 2 received additional PECS II anesthesia using 0.2% ropivacaine (volume of 20 mL). This type of anesthesia was performed solely under ultrasound guidance using the Philips Sparq ultrasound unit (USA, 2014) with a linear transducer (5–12 MHz). Basic imaging parameters (depth, gain, focus) were adjusted based on anatomical conditions to obtain the best possible image quality. The local anesthetic agent was deposited between the pectoralis major and minor muscles, blocking conduction in the lateral and medial pectoral nerves, as well as between the pectoralis minor and serratus anterior muscles, blocking the lateral branches of the intercostal nerves (Th 2–4), intercostobrachial nerve, and long thoracic nerve [4]. The anesthesia was performed using dedicated needles for ultrasound-guided blocks (Braun Ultraplex 360, Ogaki, Japan) with the length of 50 mm or 80 mm, depending on the patient’s body structure.

During the perioperative period, both groups received intravenous non-opioid analgesics. Throughout the surgical procedure, opioids, specifically fentanyl and morphine, were administered intraoperatively to manage and mitigate any signs of pain or discomfort. The decision to dispense these opioids was based on clinical signs suggestive of pain or stress during general anesthesia. These indicators included, but were not limited to, increased heart rate, elevated blood pressure, sweating, and any sudden movements or grimacing observed during the surgery. The exact dosing and choice between fentanyl and morphine was determined by the anesthetist based on the patient’s immediate needs, considering factors such as age, weight, and other coexisting conditions. After surgery, each patient was monitored and treated in the PACU and subsequently in the surgical ward.

Postoperative monitoring and pain assessment protocols, utilizing the Numerical Rating Scale (NRS), were uniform for both groups. Nursing staff systematically assessed and documented pain intensity (every 4 h or whenever the patient reported an increase in pain severity), and any instance of heightened pain (NRS > 4) was addressed with morphine administration as per patient need.

### 2.3. Data Collection

The analysis was based on data obtained from the patients’ medical records, including the anesthesia qualification form, surgical records, anesthesia records, postoperative monitoring records, physician’s orders, and assessment of pain intensity records over three postoperative days.

### 2.4. Statistical Analysis

The statistical analyses were performed using IBM SPSS Statistics 28.0 software. For quantitative variables, basic descriptive statistics were calculated with the Shapiro–Wilk test to assess the normality of the distribution. In order to compare the groups in terms of quantitative variables, the Mann–Whitney U test was utilized. For categorical data, the Pearson chi-squared test (for independence) or Fisher’s exact test (when the expected number was less than 5) was utilized. The Friedman test was conducted to compare pain intensity ratings within each patient group. A significance level of α = 0.05 was adopted for all analyses. 

## 3. Results

### 3.1. Characteristics of the Study Groups

The final analysis included data from all 200 recruited patients, with 100 individuals receiving only general anesthesia and another 100 individuals receiving the PECS 2 block in addition to general anesthesia. The statistical analysis did not reveal any significant differences between the two patient groups in terms of gender distribution, prevalence of comorbidities, ASA status, age, height, weight, BMI, duration and type of surgery. The majority of patients in both groups were female. The most common comorbidities included hypertension and obesity. The characteristics of the patient groups without the block and with the PECS II regional block, as well as the comparison between the two groups are summarized in Table 1.

### 3.2. Postoperative Pain Management during the First 72 h after Surgery

A statistically significant difference was found in the requirement for opioid analgesic medications between the two study groups. In patients without the interfascial blockage, the doses of fentanyl and morphine were higher at each evaluated stage compared to patients who received the PECS II block. Intraoperatively, the mean dose of fentanyl administered in the PECS II block group was 1.97 mcg/kg, while in the group receiving only general anesthesia, it was 3.09 mcg/kg (*p* < 0.05). The total 72-h morphine requirement also differed significantly, with an average consumption of 1.25 mg/patient in the PECS II block group, compared to 24.82 mg/patient without the block. Additionally, higher doses of metamizole were administered intraoperatively in the group without the block compared to the PECS II block group. However, the overall consumption of paracetamol and metamizole within 72 h following the surgery did not differ significantly. The effect size for these differences ranged from moderate to strong (Table 2). 

To evaluate the analgesic effectiveness of the assessed block in providing pain relief, we conducted an analysis comparing the number of patients in each group who received opioids (morphine) in the perioperative period. The number of patients receiving morphine in the group without the block was significantly higher than in the PECS II block group (93% vs. 11%) (Table 3). None of the patients who underwent the interfascial block received intraoperative morphine, and only eight patients from this group required opioid administration immediately after the surgery due to pain (NRS ≥ 4). Additionally, the necessity of introducing antiemetic medications (nausea and vomiting as an adverse effect of morphine) was analyzed. In the overall assessment of medication administration within the first 72 h after the surgery, a significantly higher percentage of patients required ondansetron and metoclopramide in the group without the additional PECS II block, owing to the adverse effects (nausea, vomiting) of opioid analgesic medications.

### 3.3. Evaluation of Pain Intensity on the Visual Analog Scale (VAS) during the Postoperative Period

Table 4 presents the comparative analysis of pain intensity assessment results between patients without the block and those with the PECS II block. Lower values on the NRS scale were statistically revealed for all stages of pain assessment in the group of patients with the PECS II block. The effect size for these differences was strong, indicating that patients with the PECS II block experienced lower pain levels immediately after anesthesia administration, in the postoperative recovery unit, upon transfer to the ward, and throughout the 3 day postoperative period (Figure 1 and Figure 2).

### 3.4. Duration of Stay in the Postanesthesia Care Unit and Length of Hospitalization after the Surgery

The length of stay in the PACU and the overall hospitalization time after the surgery were also analyzed in both study groups. It was revealed that the stay in the postoperative recovery unit was significantly shorter for patients who received the block compared to those without the block, while the overall hospitalization time after the surgery was similar for both groups (Table 5). There were no procedural complications related to the performed interfascial blockages (PECS II).

## 4. Discussion

In this prospective analysis of breast surgery patients, we primarily aimed to examine the potential advantages of PECS blocks combined with general anesthesia over general anesthesia alone. The metrics of interest included perioperative opioid consumption, postoperative pain intensity, and PACU duration. Our key findings indicated that integrating PECS 2 block resulted in significant reductions in opioid usage, enhanced pain management and a shorter PACU stay.

Introduced in 2012 by Blanco, PECS II is one of the newer ultrasound-guided interfascial regional anesthetic technique successfully adapted for breast surgery [4]. The extent of the blockade, which covers both pectoral muscles, the serratus anterior, the cutaneous innervation of the mammary gland, and partly the armpit, is optimal for ample breast surgery procedures such as mastectomy, quadrantectomy, or breast procedures extended by revision of the axillary fossa [7,8,9]. Compared to other local anesthetic techniques that can be applied in breast surgery, PECS is feasible concerning technical challenges and procedural time and is associated with fewer possible complications [7,8], as ultrasound gives a clear view of the main vessels and pleura [10,11,12,13]. One of the main results of our study is a significantly lower opioid consumption in the PECS group. Our findings are consistent with the emerging paradigm shift towards Opioid Free Anesthesia, which is built on the foundation of multimodal anesthesia. This approach acknowledges the potential detrimental effects of opioids on postoperative recovery, aiming for minimized opioid doses. Opioid-induced side effects, especially dose-dependent ones, can impede postoperative recovery, with symptoms such as nausea, vomiting, respiratory rhythm disorders, and drowsiness being among the most common [14,15,16]. Our analysis revealed that among patients without the PECS II blockade, the dose of fentanyl and morphine was higher at each stage of the procedure (intraoperatively, in the PACU and total consumption within 72 h) than in patients with the PECS II blockade (Table 2). Our results align with previous studies comparing the addition of the PECS block to general anesthesia versus general anesthesia alone [5,14,17,18,19]. Notably, morphine, the reference molecule in the panel of opioid drugs, was demanded by 93% of patients without PECS versus only 11% of those with PECS (*p* < 0.001; Table 3). The medium dose of morphine was 24.82 versus 1.25 mg, respectively, given in total within 72 h from the beginning of surgery, but mainly after recovery room discharge, which implies on-patient-demand administration. Therefore, patients received opioids mostly in non-ICU, but in less strictly monitored clinical wards, making room for delayed medical response in the case of opioid-related adverse events. Following this, the incidence of postoperative nausea and vomiting was subsequently higher in the no-PECS group, reflected in more frequent administration of antiemetic and prokinetics such as ondansetron (no-PECS 27% vs. PECS 14%, *p* = 0.023) and metoclopramide (no-PECS 14% vs. PECS 0%; *p* < 0.001) (Table 3). Apart from the aforementioned direct adverse effects of opioid administration, the actual long-term clinical impact of decreased opioid consumption in the postoperative breast surgery setting remains a matter of debate. However, effective perioperative pain management is quintessential, as inadequate pain control can predicate chronic pain development [20,21,22]. Therefore, better chronic postoperative pain management can potentially mitigate the global opioid crisis by reducing the dependence on and over-prescription of opioids, which have been a major contributing factor to opioid misuse and overdose deaths. Proper pain management can minimize prolonged opioid use, preventing the transition from short-term therapeutic use to long-term dependence and potential misuse [23,24]. Moreover, due to their immunomodulating properties, opioids may be associated with early disease progression and have a negative impact on the overall survival of cancer patients, which is specifically essential in this population [15,25,26,27,28]. Finally, considering the constant increase in breast cancer incidence the development of strategies to enhance the diagnostic procedures and treatment accessibility is mandatory. As 81% of patients diagnosed with breast cancer require surgery as part of their primary treatment, optimizing the operating room throughput is a way to increase the number of possible surgeries performed and avert the jamming of the waiting lines [29,30,31]. In our study, adding PECS to general anesthesia significantly shortened the length of stay in the PACU, which is usually one of the weakest links in postoperative care, due to the limited number of beds available (Table 5).

## 5. Conclusions

Our findings advocate for the integration of the PECS block in tandem with general anesthesia for breast cancer surgeries. The tangible benefits manifest not only in heightened analgesic efficacy but also in a marked reduction in opioid use and its consequent side effects. Given the immediate and promising outcomes from our study, there is a compelling case for incorporating PECS blocks as a routine clinical practice in breast cancer surgeries. Future studies should consider investigating extended parameters such as patient recovery speed, mental well-being, and economic implications to solidify PECS as an integral component of clinical protocol.

## Figures and Tables

**Figure 1 jpm-13-01430-f001:**
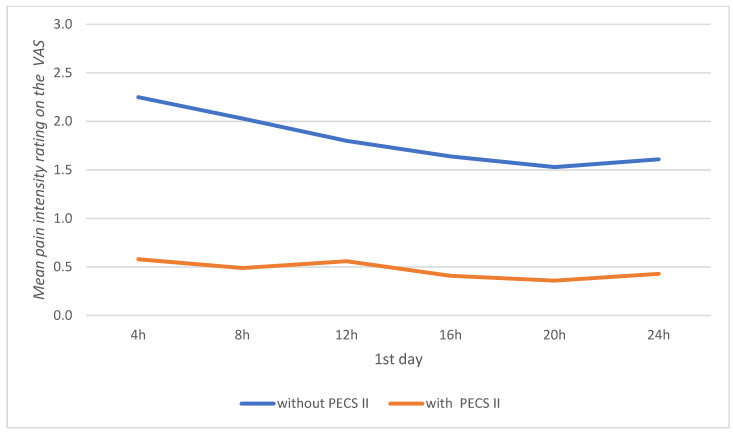
Mean pain intensity rating on the Visual Analog Scale (VAS) during the first 24 h in patients without the PECS II block and with the PECS II block.

**Figure 2 jpm-13-01430-f002:**
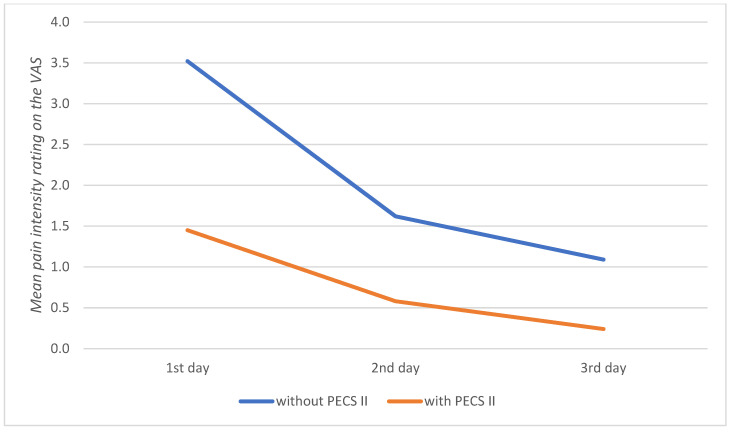
Mean pain intensity rating on the Visual Analog Scale (VAS) during the first, second, and third postoperative days in patients without the PECS II block and with the PECS II block.

**Table 1 jpm-13-01430-t001:** Characteristics and Comparative Analysis of Patient Cohorts. (without the PECS II Block and with the PECS II Block).

Patients	without PECS II (*N* = 100)	with PECS II (*N* = 100)	Test Statistic	*p*	Effect Size ^D^
Gender, *n* (%)					
Female	99 (99.0)	98 (98.0)		1.000 ^A^	0.04
Male	1 (1.0)	2 (2.0)			
Age, M (SD)	59.07 (13.24)	58.67 (14.69)	−0.13	0.900 ^C^	0.01
Height, M (SD)	161.81 (6.34)	161.57 (7.03)	−0.76	0.447 ^C^	0.05
Weight, M (SD)	74.33 (13.11)	74.52 (13.18)	−0.04	0.972 ^C^	0.00
BMI, M (SD)	28.48 (5.49)	28.56 (5.27)	−0.24	0.813 ^C^	0.02
Comorbidities, *n* (%)					
HT	36 (36.0)	49 (49.0)	3.46	0.063 ^B^	0.13
IHD	3 (3.0)	1 (1.0)		0.621 ^A^	0.07
COPD	1 (1.0)	0 (0)		1.000 ^A^	0.07
Asthma	7 (7.0)	2 (2.0)		0.170 ^A^	0.12
Diabetes	12 (12.0)	6 (6.0)		0.216 ^A^	0.10
Obesity	38 (38.0)	41 (41.0)	0.19	0.664 ^B^	0.03
CHF	6 (6.0)	1 (1.0)		0.118 ^A^	0.14
CKD	0 (0)	1 (1.0)		1.000 ^A^	0.07
ASA class, *n* (%)					
1	4 (4.0)	8 (8.0)		0.130 ^A^	0.16
2	42 (42.0)	30 (30.0)			
3	53 (53.0)	62 (62.0)			
4	1 (1.0)	0 (0)			
Surgery type, *n* (%)					
BCS	5 (5.0)	5 (5.0)	7.59	0.180 ^B^	0.19
BCS + SLNB	49 (49.0)	50 (50.0)			
Lymphadenectomy	3 (3.0)	11 (11.0)			
Simple Mastectomy	15 (15.0)	10 (10.0)			
Mastectomy + SLNB	24 (24.0)	17 (17.0)			
Mastectomy + lymphadenectomy	4 (4.0)	7 (7.0)			
Surgery duration, minutes, M (SD)	66.70 (22.57)	65.05 (29.53)	−1.20	0.231 ^C^	0.08
Surgery duration, *n* (%)					
<2 h	98 (98.0)	94 (94.0)		0.279 ^A^	0.10
2–4 h	2 (2.0)	6 (6.0)			

^A^—*p*-value for the Fisher’s Exact Test; ^B^—*p* value for the Pearson’s chi-square test; ^C^—*p*-value for the Mann–Whitney U test; ^D^—For the Pearson’s χ^2^ test and Fisher’s exact test, the measure of effect size is φ; for the Mann–Whitney U test, r is adopted. Columns not sharing a letter index differ from each other at a significance level of *p* < 0.05 (Bonferroni correction). HT—hypertension, IHD—Ischemic heart disease, COPD—chronic obstructive pulmonary disease, CHF—chronic heart failure, CKD—chronic kidney failure, BCS—breast-conserving surgery; SLNB—Sentinel lymph node biopsy.

**Table 2 jpm-13-01430-t002:** Comparison of patient groups without the PECS II block and with the PECS II block regarding doses of analgesic medications administered during the perioperative period (first 3 days).

Patients	without PECS II (*n* = 100)				with PECS II (*n* = 100)						
	Mean Rank	M	Me	IQR	Mean Rank	M	Me	IQR	*Z*	*p*	*r*
Medications administered intraoperatively											
Fentanyl (µ/kg)	135.23	3.09	3.00	1.30	65.77	1.97	2.00	0.88	−8.49	<0.001	0.60
Morphine (mg)	125.50	4.52	2.50	10.00	75.50	0.00	0.00	0.00	−8.04	<0.001	0.57
Metamizole (g)	110.28	1.92	2.00	1.00	77.66	1.57	1.50	0.00	−4.94	<0.001	0.36
Paracetamol (g)	36.50	1.00	1.00	0.00	36.50	1.00	1.00	0.00	0.00	1.000	0.00
Medications administered in PACU											
Morphine (mg)	111.08	2.20	0.00	4.00	89.92	0.78	0.00	0.00	−3.78	<0.001	0.27
Medications administered cumulatively within 72 h											
Fentanyl (µ/kg)	135.23	3.09	3.00	1.30	65.77	1.97	2.00	0.88	−8.49	<0.001	0.60
Morphine (mg)	144.36	24.82	24.00	27.75	56.65	1.25	0.00	0.00	−11.36	<0.001	0.80
Metamizole (g)	94.37	9.06	9.00	4.75	101.60	9.47	9.00	4.88	−0.90	0.368	0.06
Paracetamol (g)	93.96	7.71	8.00	4.00	95.93	7.87	8.00	4.75	−0.25	0.803	0.02

µ/kg—microgram per kilogram of body weight; mg—milligram, g—gram; PACU—postanesthesia care unit.

**Table 3 jpm-13-01430-t003:** Comparison of pain medication demand between the group without the PECS II block and the group with the PECS II block.

Patients	Without PECS II		With PECS II				
	*N*	*%*	*N*	*%*	χ^2^	*p*	φ
Medications administered intraoperatively							
Morphine	50	50.0	0	0		<0.001	0.58
Metamizole	87	87.0	98	98.0	8.72	0.003	0.21
Paracetamol	66	66.0	6	6.0	78.12	<0.001	0.62
Medications administered in PACU							
Morphine	30	30.0	8	8.0	15.72	<0.001	0.28
Medications administered cumulatively within 72 h							
Morphine	93	93.0	11	11.0	134.70	<0.001	0.82
Metamizole	97	97.0	98	98.0		1.000	0.03
Paracetamol	89	89.0	100	100.0	11.64	<0.001	0.24
Ondansetron	27	27.0	14	14.0	5.18	0.023	0.16
Metoclopramide	14	14.0	0	0		<0.001	0.27

χ^2^—Pearson’s chi-square statistic; φ—Fisher’s exact test; PACU—postanesthesia care unit.

**Table 4 jpm-13-01430-t004:** Comparison of pain intensity ratings on the Numeric Rating Scale (NRS) between patients without the PECS II block and those with the PECS II block.

Patients	without PECS II (*n* = 100)				with PECS II (*n* = 100)						
	Mean Rank	M	Me	IQR	Mean Rank	M	Me	IQR	*Z*	*p*	*r*
Immediately after anesthesia administration.	147.12	2.05	2.00	2.00	53.88	0.11	0.00	0.00	−12.17	<0.001	0.86
PACU	142.54	3.25	3.00	3.00	58.47	0.59	0.00	0.00	−10.64	<0.001	0.75
PostPACU discharge	144.41	2.37	2.00	1.00	56.60	0.48	0.00	1.00	−11.22	<0.001	0.79
Postoperative timeline											
4 h	141.08	2.25	2.00	1.00	59.93	0.58	0.00	1.00	−10.26	<0.001	0.73
8 h	140.62	2.03	2.00	1.75	60.38	0.49	0.00	1.00	−10.23	<0.001	0.72
12 h	136.44	1.80	2.00	1.00	64.57	0.56	0.00	1.00	−9.21	<0.001	0.65
16 h	137.04	1.64	2.00	1.00	63.96	0.41	0.00	1.00	−9.41	<0.001	0.67
20 h	138.93	1.53	2.00	1.00	62.08	0.36	0.00	1.00	−9.96	<0.001	0.70
24 h	136.98	1.61	2.00	1.00	64.02	0.43	0.00	1.00	−9.43	<0.001	0.67
Highest pain score recorded											
1st postoperative day	138.23	3.52	3.00	3.00	62.78	1.45	1.00	1.00	−9.46	<0.001	0.67
2nd postoperative day	135.13	1.62	1.00	1.00	65.88	0.58	1.00	1.00	−9.15	<0.001	0.65
3rd postoperative day	132.24	1.09	1.00	0.00	68.77	0.24	0.00	0.00	−8.58	<0.001	0.61

PACU—postanesthesia care unit.

**Table 5 jpm-13-01430-t005:** Comparison of the duration of stay in the postanesthesia care unit and length of hospitalization following surgery in patients without the PECS II block and those with the PECS II block.

	without PECS II (n = 100)				with PECS II (n = 100)						
	Mean Rank	M	Me	IQR	Mean Rank	M	Me	IQR	*Z*	*p*	*r*
Duration of stay in PACU (h)	119.40	2.44	2.00	1.50	81.61	1.47	1.00	1.00	−5.02	<0.001	0.35
Length of hospitalization following surgery.	102.46	4.24	4.00	2.75	98.55	3.99	4.00	2.00	−0.49	0.622	0.03

PACU—postanesthesia care unit.

## Data Availability

Resource data are available upon request.

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
