# Peer review of "Evaluation of the Efficacy of Pectoral Nerve-2 Block (PECS 2) in Breast Cancer Surgery"

_jpm, 2023, doi:10.3390/jpm13101430_

Round 1

Reviewer 1 Report

dear authors,

It is an interesting topic however, some topics must be improved.

Introdution: must be improved, explaining the importance of this topic and diescrebed with more detail the method used and the condicionants of brest surgery

Methods: must be more precise on method description: eg. inclusion criteria

the conclusions must be improved, with some clinical importance of this study and implementation

Author Response

Dear Reviewer, 

We appreciate your constructive feedback on our manuscript. We have carefully addressed your comments and suggestions, incorporating significant improvements into the manuscript. Below, we outline the key revisions made in response to your valuable input. All implemented changes in the manuscript are highlighted in yellow. 

1. **Improved Introduction Section:**
   In response to your feedback, we have enhanced the introduction section to better emphasize the critical importance of advancing breast cancer treatment. We have incorporated additional context and background information that underscores the significance of our study within the broader context of breast cancer management. These revisions aim to provide a clearer understanding of the overarching importance of our research.

2. **Revised Methods Section:**
  We have thoroughly revised the methods section to provide a more detailed and transparent account of the patient inclusion criteria used in our study. This improved description ensures that readers can readily grasp the selection process for participants. We have incorporated specific criteria, justifications, and ethical considerations to enhance the clarity and completeness of this section.

3. **Revised Conclusion:**
  The conclusion section has been reworked to better summarize the clinical importance of our study. We have refined the language to emphasize the clinical significance of our findings, particularly in relation to breast cancer surgery. The revised conclusion highlights how the integration of the PECS block into anesthesia protocols can improve patient outcomes and contribute to enhanced postoperative care, including potential reductions in hospital stays and improved operative room efficiency.

We believe that these revisions significantly strengthen the manuscript. 

Thank you for your time and consideration. We look forward to your feedback on these revisions and hope that our manuscript is now better poised for publication.

Sincerely,

Malgorzata Celinska - Spodar 

Reviewer 2 Report

This is a well-designed and well-presented work. Congratulations on the quality of presentation, it is a manuscript for experts, however everyone can understand it. The sample of patients was appropriate, proven by all statistical important conclusions.Demographic data, operative data, patient satisfaction and pain managment, are all adequaly presented. A manuscript well structured and fluent. Well done!

Author Response

Dear Reviewer

We would like to express our sincere gratitude for your thorough review of our manuscript. We are delighted to receive your feedback and your positive assessment of our work.

Once again, thank you for your time and encouragement.

We look forward to the opportunity for our research to reach a wider audience and contribute positively to the field.

Sincerely,

Malgorzata Celinska-Spodar